# Imaging the electron charge density in monolayer MoS₂ at the Ångstrom scale

Joel Martis[1,12], Sandhya Susarla[2,3,4,12], Archith Rayabharam [5], Cong Su [6,7,8], Timothy Paule[6,7,8], Philipp Pelz[2,9], Cassandra Huff[10], Xintong Xu [1], Hao-Kun Li[1], Marc Jaikissoon[10], Victoria Chen[10], Eric Pop [10], Krishna Saraswat [10], Alex Zettl [6,7,8], Narayana R. Aluru [11], Ramamoorthy Ramesh [6,7], Peter Ercius [2] ✉ & Arun Majumdar [1] ✉

Four-dimensional scanning transmission electron microscopy (4D-STEM) has recently gained widespread attention for its ability to image atomic electric fields with sub-Ångstrom spatial resolution. These electric field maps represent the integrated effect of the nucleus, core electrons and valence electrons, and separating their contributions is non-trivial. In this paper, we utilized simultaneously acquired 4D-STEM center of mass (CoM) images and annular dark field (ADF) images to determine the projected electron charge density in monolayer MoS₂. We evaluate the contributions of both the core electrons and the valence electrons to the derived electron charge density; however, due to blurring by the probe shape, the valence electron contribution forms a nearly featureless background while most of the spatial modulation comes from the core electrons. Our findings highlight the importance of probe shape in interpreting charge densities derived from 4D-STEM and the need for smaller electron probes.

Four-dimensional scanning transmission electron microscopy (4D-STEM) has become a versatile tool in recent years with applications ranging from measuring nanoscale strain to uncovering thermal vibrations of atoms[1,2]. One such 4D-STEM technique measures local electric fields by calculating the center of mass (CoM) of the diffraction pattern[3]. In the past few years, sub-Ångstrom electric field and charge density mapping using 4D-STEM CoM imaging has become feasible due to aberration-corrected STEMs and fast pixelated detectors[4–9]. Atomic electric fields emerge from a combination of strong nuclear effects and weak valence electrons that form chemical bonds. The ability to map valence electrons with high spatial resolution can potentially lead to new insights about chemical bonding, charge transfer effects, polarization, ferroelectricity, ion transport, and much more[10,11].

Imaging valence electrons at the atomic scale is a non-trivial problem. Annular dark field (ADF) STEM, for example, images atom positions based on the high-angle scattering of incident electrons by the nucleus[12,13]. Phase contrast high resolution (HR-) TEM can reveal chemical bonding effects due to charge redistribution, but electron orbital charge densities have not been explicitly imaged[14]. Electron

[1]Department of Mechanical Engineering, Stanford University, Stanford, CA, USA. [2]The National Center for Electron Microscopy (NCEM), The Molecular Foundry, Lawrence Berkeley National Laboratory, Berkeley, CA, USA. [3]Materials Science Division, Lawrence Berkeley National Laboratory, Berkeley, CA, USA. [4]School for Engineering of Matter, Transport and Energy, Arizona State University, Tempe, AZ, USA. [5]Department of Mechanical Engineering, University of Illinois at Urbana-Champaign, Urbana, IL, USA. [6]Department of Physics, University of California Berkeley, Berkeley, CA, USA. [7]Department of Materials Science and Engineering, University of California Berkeley, Berkeley, CA, USA. [8]Kavli Energy NanoScience Institute, University of California Berkeley, Berkeley, CA, USA. [9]Institute of Micro- and Nanostructure Research & Center for Nanoanalysis and Electron Microscopy (CENEM), Department of Materials Science, Friedrich-Alexander-Universität Erlangen-Nürnberg (FAU), Erlangen, Germany. [10]Department of Electrical Engineering, Stanford University, Stanford, CA, USA. [11]Department of Mechanical Engineering, The University of Texas at Austin, Austin, TX, USA. [12]These authors contributed equally: Joel Martis, Sandhya Susarla. ✉e-mail: percius@lbl.gov; amajumdar@stanford.edu

holography can yield atomic scale potentials and charge densities; however, the nuclear and electronic effects are non-trivial to separate and electron orbitals haven't been explicitly imaged[15]. Core-loss electron energy loss spectroscopy (EELS) can identify core electron states at atomic resolution[16] but cannot measure their charge density directly. Valence EELS (VEELS) is limited by the delocalization of the excitation on the nanometer scale, much larger than the size of the valence orbitals themselves[17]. Although recent VEELS work has shown atomic-scale contrast in certain energy ranges in graphene, the contrast is a function of inelastic scattering cross sections between different orbitals and sample thickness, making it non-trivial to isolate valence electron charge densities[18,19]. Valence electron densities are commonly measured using scanning tunneling microscopy (STM)[20], but these are limited to surfaces and energy ranges typically only a few eV below the Fermi level[21]. While previous efforts have shown that electron contributions are important in 4D-STEM images[4,8], the electron charge density has not been explicitly imaged so far.

In this paper, we use monolayer two-dimensional 2H-MoS$_2$ as a model system to investigate the contributions of atomic electric fields and charge densities in a 4D-STEM dataset. In particular, we show how the ADF-STEM intensity channel can be used to subtract the nuclear contribution from the total charge density derived from 4D-STEM and derive the electron charge density in MoS$_2$. Our experimental results show good agreement with the electron charge density predicted by density functional theory (DFT). We discuss how both core electrons and valence electrons contribute to the derived electron charge density, and how probe convolution (i.e., blurring by the incident probe intensity distribution) results in core electrons dominating the measured electron charge density map. We also discuss how residual aberrations in the instrument can have a sizeable impact on the charge density image. Our findings point towards a need for smaller electron probes and precise probe deconvolution methods that could potentially distinguish between valence and core electrons based on orbital size.

## Results

### 4D-STEM of monolayer MoS$_2$

A 4D-STEM dataset is acquired by scanning a focused electron probe across a sample and using a pixelated detector to image the scattered electron beam at each probe position (Fig. 1a). It has been shown using Ehrenfest's theorem that the CoM of the scattered electron beam at each probe position is directly proportional to the projected electric field at that probe position convolved with the probe intensity[3]. Therefore, one can derive a 2D electric field map of a sample by simply computing the CoM of the scattered electron beam at every probe position as it scans across a sample. This electric field map can then be converted into a projected charge density image and an electrostatic potential image of the sample using Gauss' law.

Here, we derive atomic electric field maps of monolayer MoS$_2$ using 4D-STEM CoM imaging. A monolayer of MoS$_2$ is a two-dimensional direct bandgap semiconductor in its 2H phase where the Mo atoms are sandwiched between two S atoms (Fig. 1b). The semiconducting nature and direct band gap are useful for optoelectronics and catalysis applications[18,19]. Fig. 1c shows an ADF-STEM image of a super-cell of MoS$_2$. Simultaneously, the transmitted beam intensity is imaged using a high speed 4D-STEM camera[22], and the CoM of the diffraction pattern at each probe position is computed, leading to Fig. 1d, e. The camera is a direct electron detector and allows for high quantum efficiency data collection at high speeds, which is critical when imaging beam sensitive materials such as monolayer MoS$_2$. Fig. 1c–e represents unit cell averages over about 25 super-cells from a larger scan area which significantly improves the signal-to-noise ratio (SNR) (see Supplementary Fig. 3).

Since the CoM of the transmitted electron beam in each diffraction pattern is proportional to the projected electric field at the sample, the experimental projected electric field map in Fig. 2a is derived by simply multiplying the CoM images with appropriate physical constants, following ref. 3. The intensities of the image pixels represent the magnitude of the electric field, and the arrows represent its direction. We observe that the centers of lattice sites, midpoints

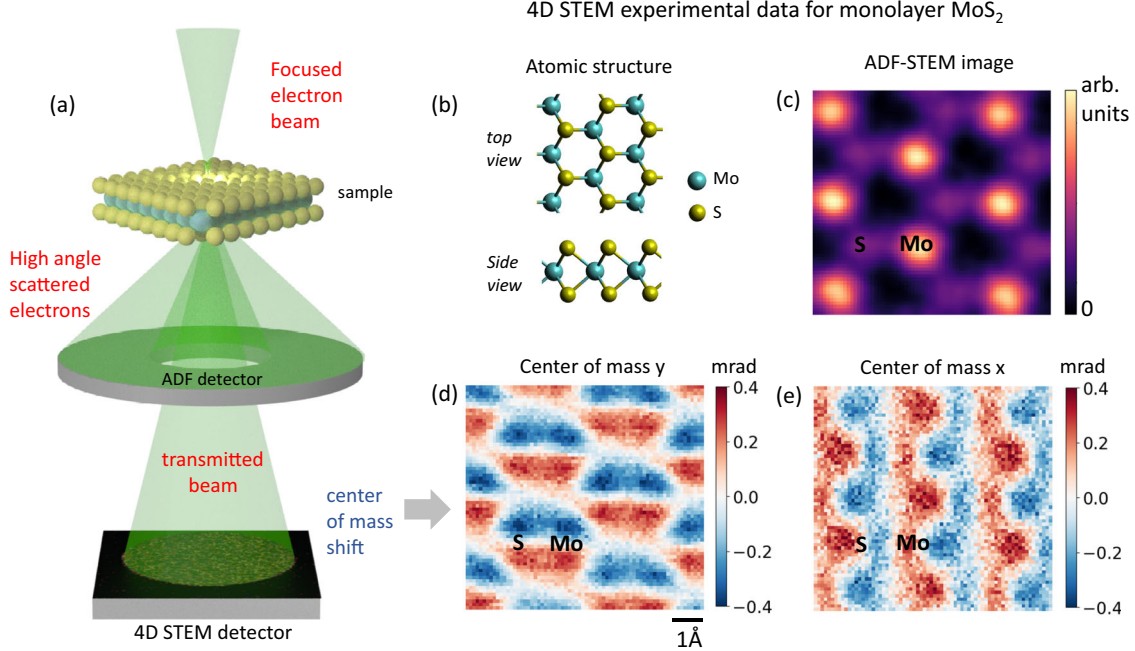

**4D STEM experimental data for monolayer MoS$_2$**

Atomic structure — ADF-STEM image — Center of mass y — Center of mass x

**Fig. 1 | Four-dimensional scanning transmission electron microscopy (4D-STEM) experiment on monolayer MoS$_2$.** **a** Schematic showing the experimental setup for simultaneous 4D-STEM and Annular Dark Field (ADF)-STEM imaging. **b** Atomic structure of a super-cell of MoS$_2$. **c** Super-cell averaged ADF-STEM image. **d** Center of mass along y and (**e**) center of mass along x corresponding to (**c**). All images have the same scale with the scale bar in (**d**).

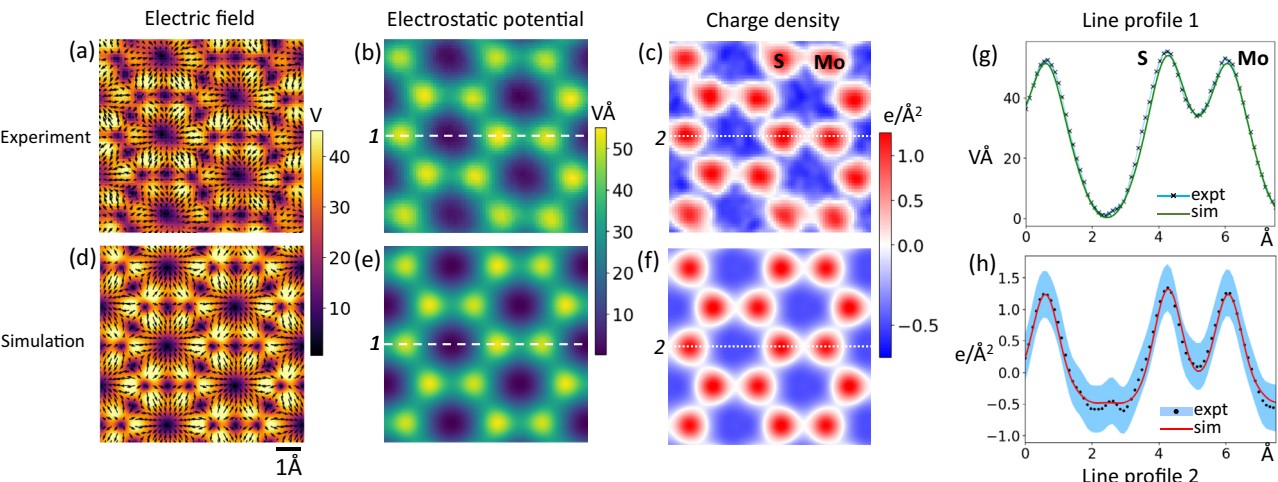

**Fig. 2 | Electric field and charge density maps from Center of Mass (CoM) imaging. a** Experimental electric field map corresponding to a unit cell. The color intensity indicates the magnitude and arrows indicate the direction of the electric field. **b** Electrostatic potential and (**c**) charge density corresponding to (**a**) with red indicating net positive charge and blue indicating net negative charge. **d** Simulated electric field map from Density Functional Theory (DFT) and resulting (**e**) electrostatic potential and (**f**) charge density convolved with an electron probe at 80 keV and 30 mrad convergence semiangle. Line profiles of the (**g**) electrostatic potential and (**h**) charge density comparing (**b**, **c**) experiment with (**e**, **f**) simulation. The light blue shaded regions indicate one standard deviation on either side of the mean of the experimental data. All images share the same scale with the scale bar in (**d**).

between neighboring atoms, and the centers of the hexagonal cells show zero electric field in agreement with previously reported works[7,9]. Using the projected electric field, we computed the projected potential by integrating the field (Fig. 2b) and the projected charge density using Gauss' Law (Fig. 2c). To further improve the SNR in the charge density image, we use a Gaussian filter with a 0.4 Å FWHM. The projected charge density (Fig. 2c) is the sum of the nuclear and electron charge densities convolved with the probe intensity. The lattice sites have a net positive charge (red) because of a higher contribution from the nucleus, whereas the regions around lattice sites have a net negative charge (blue) indicating a higher contribution from the electrons. It is non-trivial to separate the contributions of the nucleus and the electrons to the net charge density and the contributions of valence and core electrons to the electronic part of the net charge density.

**Effect of probe convolution**

The incident electron probe's shape is an Airy function with a central peak of ~1 Å in size and so-called probe tails that extend farther beyond the central peak[23]. Thus, probe convolution plays a fundamental role in interpreting Fig. 2a–c. Extended probe tails arise because of geometrical and chromatic aberrations in the probe. Chromatic aberrations focus electrons with different energies to different points along the optic axis, giving rise to a 'focal spread' in the probe. In (spherical) aberration corrected STEMs, residual geometrical aberrations due to corrector alignment drift and measurement error also play a significant role in determining the extended shape of the probe[24]. We simulate the 4D-STEM data by convolving the projected electric field, potential and charge density derived from DFT with the electron probe intensity calculated for 80 kV and 30 mrad convergence semi-angle along with chromatic and residual geometrical aberrations (−1 nm defocus, −10 nm threefold astigmatism, 5 μm 3rd order spherical aberration, 7.5 nm FWHM chromatic focal spread) (Fig. 2d–f). These figures also include a 0.7 Å FWHM Gaussian blurring to account for the source size and other blurring factors. Monolayer $MoS_2$ is sufficiently thin that the use of the projected potential approximation instead of full multislice calculations is justified (see Supplementary Fig. 6). Reasonable residual probe aberrations were determined empirically to match experimental data and are well within reasonable limits of microscope performance during extended operation. The

contributions to the charge density image arising from different aberrations is illustrated in Fig. 3, where the top row shows the probe shape, and the bottom row shows the corresponding probe convolved charge density image. Many different combinations of aberrations could lead to a similar probe shape, and all we demonstrate here is the effect of probe tails on such experimental images. An ideal diffraction limited probe at 30 mrad gives rise to a charge density image that is about 4 times higher in magnitude compared to our experimental data (Fig. 3a, f). Adding 7.5 nm of chromatic focal spread (FWHM, corresponding to 0.6 mm Cc, 1 eV energy spread at 80 keV), the probe tails become more prominent resulting in diminished image intensity (Fig. 3b, g). Adding 5 μm of 3rd order spherical aberration ($C_{30}$ using Krivanek's notation[25]), we see a further reduction in image intensity (Fig. 3c, h). Adding −1 nm of defocus (denoted as -$C_{10}$) and −10 nm threefold astigmatism ($C_{23}$), the simulated charge density image begins to resemble the experiment (Fig. 3d, i). To illustrate the out-sized impact of residual aberrations, we rotate the threefold astigmatism by 180° and increase it to 15 nm. The resulting image shows different peak intensities on atomic sites (higher S intensity than Mo) and a different charge density distribution between atoms (Fig. 3e, j). Our analysis helps explain two important issues. (1) The charge density image intensities observed in experiments (ours and others[7,9,26]) are lower than predicted by ideal (aberration free) simulations likely because of intensity redistribution in the probe tails due to chromatic and residual geometrical aberrations. (2) Even small residual aberrations can change the apparent charge density distribution giving rise to artefacts. Residual aberrations are difficult to quantify fully since they may not directly affect the point resolution of the structural image which is largely determined by the full width half maximum of the probe's central peak. However, they can have an outsized impact on the extended intensity distribution in the charge density image and must be taken into account before making conclusions about the distribution of electrons around atoms.

To understand the effect of probe convolution on the electron charge density, we convolve the DFT-derived electron charge density with the simulated probe shape as determined earlier (Fig. 4). Fig. 4a, b shows the DFT simulated valence and core electron charge densities for $MoS_2$. The valence electrons consist of the $3s^2 3p^4$ electrons from S and the $4d^5 6s^1$ from Mo. We observe that the valence electrons are predominantly concentrated on the S atom (12 e- from the 2 Sulfur

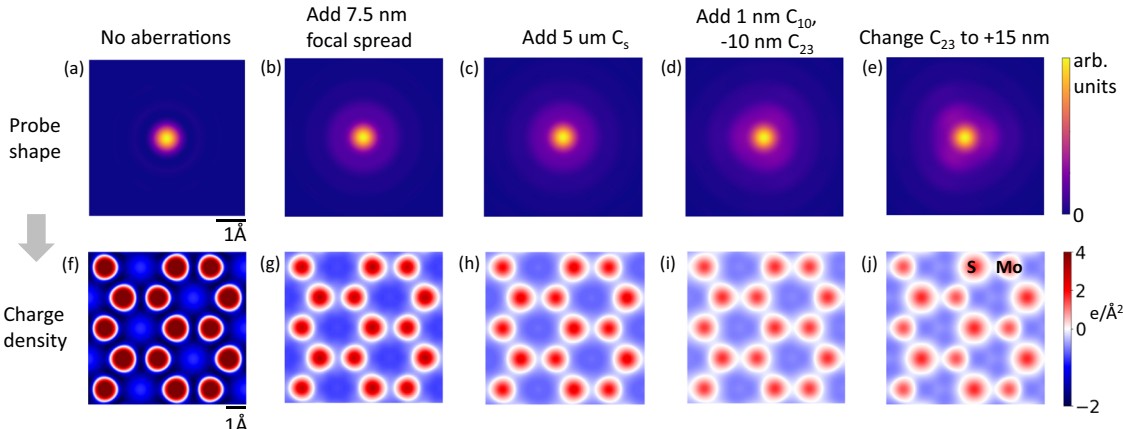

**Fig. 3 | Effect of probe shape and residual aberrations on the charge density image.** Simulated probe shapes (**a**–**e**) and net charge density images (**f**–**j**) while successively adding various aberrations: **a**, **f** a 30 mrad probe at 80 kV with no chromatic or geometric aberrations. **b**, **g** Adding chromatic aberrations (7.5 nm FWHM focal spread). **c**, **h** Adding 5 μm 3rd order spherical aberrations, Cs (also known as $C_{30}$). **d**, **i** Adding 1 nm $C_{10}$ (negative defocus), −10 nm $C_{23}$ (threefold astigmatism). **e**, **j** Changing $C_{23}$ to +15 nm. Images (**a**–**e**) share the same scale bar as in (**a**), and images (**f**–**j**) share the same scale bar as in (**f**).

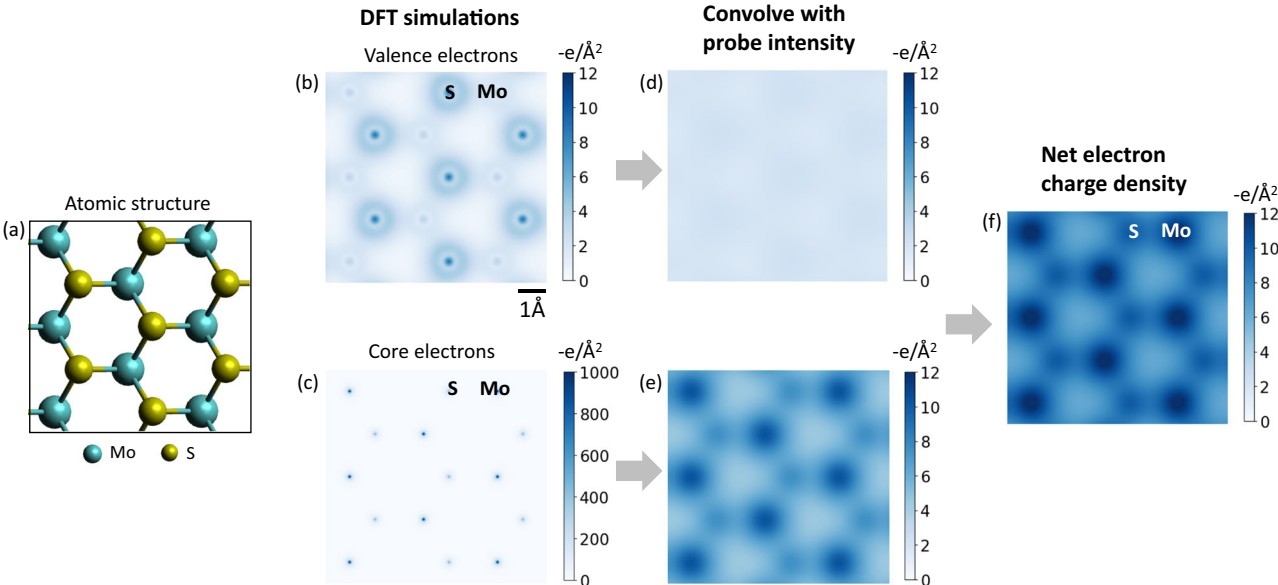

**Fig. 4 | Simulated contributions of valence and core electrons to the electron charge density. a** Atomic structure of an $MoS_2$ supercell. **b** DFT derived valence electrons and (**c**) core electrons corresponding to the atomic structure in (**a**). Probe convolved with (**d**) valence and (**e**) core electrons. **f** Sum of (**d**) and (**e**) showing the net electron charge density. All images share the same scalebar with (**b**).

atoms stacked on top of each other, compared to 6 e- from Mo). The core electron charge density shows a higher charge density on Mo as expected. Probe convolution (Fig. 4d, e) shows that the core electron charge density is delocalized on the order of 1 Å and the valence electron charge density is nearly featureless. Their sum, Fig. 4f, produces the net electron charge density where any variations mostly come from the core electrons. The valence electrons contribute a uniform background of about −2.5 e/Å².

### Imaging the electron charge density
To isolate the electron charge density experimentally, the nuclear charge density needs to be subtracted from the total charge density derived using 4D-STEM CoM imaging. Because the nucleus, which is confined to a few femtometers in size, scatters electrons to much higher angles relative to core electrons and valence electrons, it predominantly contributes to ADF-STEM image contrast. The nuclear

scattering is proportional to $Z^x$, where Z is the atomic number and the exponent x ranges from 1.5–2 depending on the collection geometry, probe aberrations and the Z of the studied elements[12,27,28]. In our experiments, we find that the ADF-STEM intensity for S and Mo scales as $\sim Z^{1.7}$ which we validate with multislice simulations using abTEM[29] (see Supplementary Note 1 and Supplementary Fig. 5). Using prior knowledge that the sample is 1 layer of $MoS_2$, we assign each experimentally determined atom position a discrete nuclear charge. Ideally, one could calibrate a given ADF-STEM detector with several types of atoms and derive a direct correlation between observed ADF-STEM intensity and quantitative nuclear charge, as done in[12]. This could allow for direct quantification of the nuclear charge of atomic columns of unknown composition and thickness, which is unnecessary in the case of our sample. Once a nuclear charge is assigned as a delta function, we convolve it with the aberrated probe intensity along with 0.7 Å FWHM Gaussian blurring to account for source size (see Supplementary

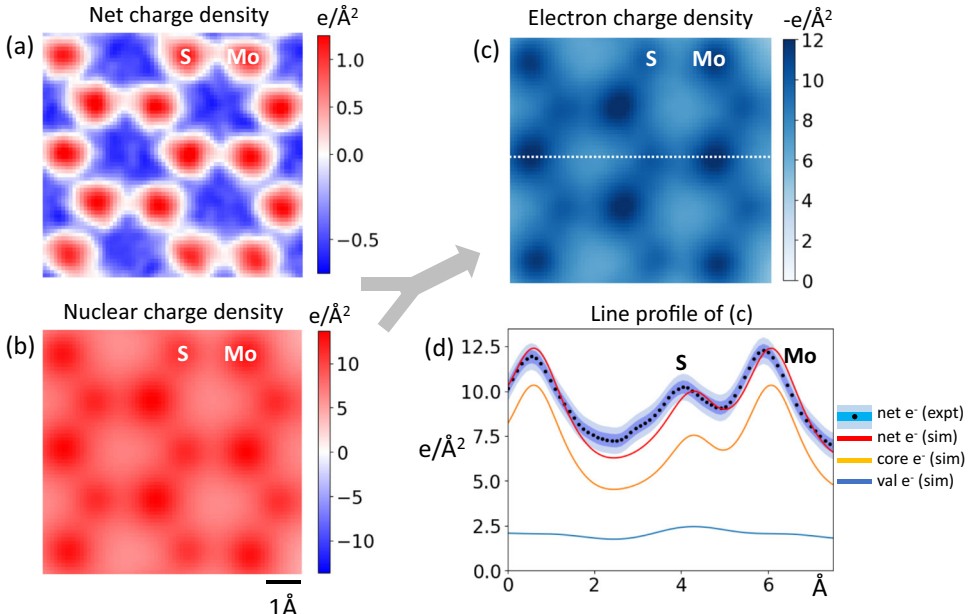

**Fig. 5 | Deriving the electron charge density from experimental data.**
**a** Experimental electron charge density. **b** Reconstructed nuclear charge density with probe convolution. **c** Experimental electron charge density derived by subtracting (**b**) from (**a**). **d** Line profiles for the experimental electron charge density and simulated charge density from Fig. 4f, showing the contributions of valence and core electrons to the overall electron charge density. The light and dark blue shaded regions indicate one and two standard deviation(s) on either side of the mean, respectively. All images share the same scalebar with (**b**).

Fig. 5). Fig. 5b shows the nuclear charge density image convolved with the probe intensity, which when subtracted from the net charge in Fig. 5a, gives the electron charge density in Fig. 5c. Fig. 5d shows the line profiles comparing the experimentally derived and simulated electron charge densities. We observe that the spatial modulation in the experimental electron charge density comes from the core electrons, with the valence electrons contributing as a uniform background. This means that even though CoM imaging can image electron distributions, probe convolution blurs the valence electron density forming an almost uniform, unmodulated background of about $-2.5$ e/Å². We note that the experimental noise (std $\sigma \sim 0.4$ e/Å²) is much lower than valence electron density, indicating that we can indeed measure the effect of the valence electron charge density.

## Discussion

We have imaged the electron charge density in monolayer $MoS_2$ using 4D-STEM CoM imaging and explored the contributions from the valence v/s core electrons. We found that probe convolution smooths out any features in the valence electron charge density, and the spatial modulation in the derived electron charge density mostly comes from core electrons. Our findings highlight the importance of probe shape in quantitatively interpreting CoM-derived charge density images. Residual aberrations that extend probe tails can significantly diminish image intensities, and asymmetry in the probe shape can give rise to asymmetric charge densities. Residual aberrations in the probe must therefore be accounted for when interpreting the spatial distribution of charge densities in CoM 4D-STEM results.

Because valence electron orbitals are of interest for understanding the chemical and electronic properties of materials, an important question to address is – can the valence electron orbitals be imaged from CoM images? In order to separate valence and core electron charge densities, one needs an electron probe size that is much smaller than the 1 Å feature size of valence electron orbitals. Achieving this experimentally is currently challenging as state-of-the-art probes are on the order of valence electron orbital sizes (about 0.7 Å at 80 keV and 0.5 Å at 300 keV). Some improvement in valence

electron contrast can be achieved by eliminating effects of geometric and chromatic aberrations (compare Fig. 6a–c to d–f). Increasing the convergence angle by 2× to 60 mrad (i.e., decreasing the probe size by 2×) also shows an improvement in separation between valence and core electrons, the latter being localized around atomic sites (Fig. 6g–i). The exact requirements for quantitatively imaging valence electron densities depend on the material being studied and SNR achievable without damaging the material in addition to a well characterized and aberration free probe. With a sufficiently small and well characterized probe, we expect CoM imaging to yield unique insights into chemical bonding in thin materials (few nm) where the phase object approximation is valid. Materials with internal dipoles such as ferroelectrics will be particularly interesting to study, and a small aberration free probe might allow for quantitative characterization of the dipole moment within a unit cell. Further reducing the probe size requires non-trivial solutions that can mitigate higher order aberrations[30], chromatic effects and other limitations such as Johnson noise[31].

Post-processing reconstruction methods such as ptychography offer an indirect route to attaining super-resolution images by iteratively refining and disentangling nuisance parameters such as position errors and partial coherence[2,32–34]. Spatial resolution in electron microscopy at the atomic scale is typically quantified by two approaches. Self-consistency measures like Fourier Ring correlation (FRC)[35] or spectral signal-to-noise ratio (SSNR)[36] measure the consistency of a reconstruction from two experimental datasets. If single atoms or atomic columns are visible, optical resolution can be measured by the minimum resolvable separation between atoms[34]. Since the contribution of valence electrons to the charge density is roughly two orders of magnitude lower than the contribution of core electrons, it will be critical to evaluate the sensitivity of these existing resolution metrics regarding quantitative phase reconstruction. We emphasize here that although CoM imaging is affected by probe shape it offers a more direct route to imaging charge densities, as it simply relies on calculating the divergence of the center of mass of the transmitted electron beam. Our findings point towards the need for smaller electron probes and better

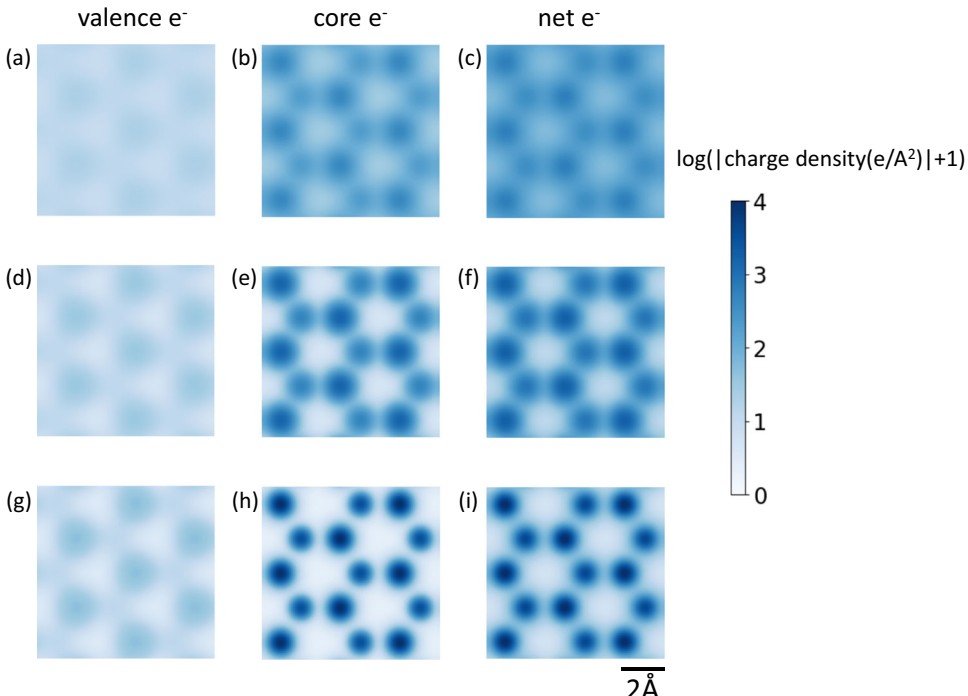

**Fig. 6 | Simulated valence and core electron charge densities for smaller probes.** Valence, core, and net electron charge densities for a geometrical aberration free 80 kV electron beam with: **a**–**c** a 30 mrad convergence angle with 7.5 nm FWHM chromatic focal spread and 0.7 Å source size. **d**–**f** 30 mrad convergence angle and no chromatic and 0.7 Å source size. **g**–**i** 60 mrad convergence angle and no chromatic and 0.35 Å source size. Note that all images are shown with a logarithmic intensity scaling calculated as $\log_e(|\text{charge density}|+1)$. All images share the same scalebar with (**i**).

characterization and control of the probe shape which could potentially resolve valence electron charge densities in materials.

## Methods

### Sample preparation

Monolayer $MoS_2$ is grown by the halide-assisted chemical vapor deposition method with a Mo source provided by a liquid precursor. The detailed procedure is described in[37]. We subsequently transfer the monolayer $MoS_2$ samples to a Quantifoil TEM grid (2/2, copper grid) by the standard wet transfer method using PMMA which is also described in[37]. The PMMA is cleaned by acetone vapor before imaging in STEM. Optical images of $MoS_2$ before and after transfer are shown in Supplementary Fig. 1.

### 4D STEM experiments

The 4D-STEM data was acquired using a Titan 80–300 called the TEAM 0.5 at the National Center for Electron Microscopy facility of the Molecular Foundry. The machine was operated in STEM mode at 80 kV with a convergence semi-angle of 30 mrad and approximately 20 pA beam current. A direct electron detector called the 4D Camera was used to capture a diffraction pattern at each probe position. The camera has 576 × 576 pixels and operates at a frame rate of 87,000 Hz. The full data set consists of 512 × 512 probe locations and 4 frames were summed at each position (Supplementary Fig. 2). The dose used per probe position was ~$10^6$ e/$Å^2$.

Unit cell averaging was used to improve the SNR ratio (Supplementary Fig. 3). This was done using three steps: (1) Atom positions were identified in the electrostatic potential image (since it was the less noisy compared to ADF and CoM images) using AtomSegNet[38], a deep learning based localization algorithm. (2) Atoms were classified into Mo and S based on the simultaneously acquired ADF-STEM image intensity. (3) Using atom positions, super-cell averaging was carried out at each atom site in the CoM, phase and charge density images to yield unit cell averaged images. We observed a significant

improvement in the SNR of the CoM and charge density images after averaging.

### DFT Simulations

To simulate the ground state charge density distribution of the 2H-phase of $MoS_2$, a self-consistent analysis of the Density Functional Theory (DFT)[39] was performed using the Vienna Ab initio simulation (VASP)[40,41] package. The Perdew-Burke-Ernzerhof (PBE)[42] exchange-correlation functional, which comes under the Generalized Gradient Approximation (GGA), was used and projected augmented wave (PAW) pseudopotentials with a 500 eV energy cutoff and Gamma-point-centered k-point of 8 × 8 × 2 were used. The convergence with respect to the grid-size and cut-off energy is shown in Supplementary Fig. 4 for a sample simulation. The unit cell of $MoS_2$ used consists of 12 atoms, with the simulation box having the dimensions 6.325 × 5.478 × 12.302 $Å^3$. Before the self-consistent calculation for evaluating the charge density distribution are done, the 2H-phase of $MoS_2$ is relaxed using an energy convergence criteria of $10^{-8}$ eV.

### STEM simulations

STEM probe intensities were calculated using methods outlined in[23] using in-house python code. A 30 mrad aperture was used along with residual aberrations mentioned in the manuscript. 4D-STEM CoM charge densities and electric fields were simulated by convolving the simulated probe intensity with the DFT simulated electron charge density and manually added delta functions to represent nuclear charge densities. The nuclear charge image in Fig. 5 was generated by convolving a probe shape with discrete nuclear charge densities placed at atom positions. Atom positions were determined as either Mo or S based on the intensity in the simultaneously acquired ADF-STEM image. Multislice ADF-STEM simulations were carried out using abTEM. The details are described in Supplementary Note 1 and Supplementary Fig. 5. Simulations in Supplementary Fig. 6 were carried out using abTEM.

## Data availability

The data that support the findings in this study are available at Zenodo (https://doi.org/10.5281/zenodo.7916671).

## Code availability

The code that support the findings in this study is available at Zenodo (https://doi.org/10.5281/zenodo.7916671).

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

## Acknowledgements

J.M. and A.M. acknowledge support from the Air Force Office of Scientific Research under Grant No. FA9550-19-1-0309. J.M., X.X. and A.M. were supported as part of the Center for Enhanced Nanofluidic Transport (CENT), an Energy Frontier Research Center funded by the U.S. Department of Energy (DOE), Office of Science, Basic Energy Sciences

(BES), under Award No. DE-SC0019112. S.S and R.R. were supported by the DOE Office of Science, Basic Energy Sciences, Materials Sciences, and Engineering Division under contract DE-AC02-05-CH11231 within the Quantum Materials program (KC2202). The experiments were performed at the Molecular Foundry, Lawrence Berkeley National Laboratory, which is supported by the U.S. Department of Energy under contract no. DE-AC02-05CH11231. We would like to thank Gatan, Inc. as well as P Denes, A Minor, J Ciston, C Ophus, J Joseph, and I Johnson who contributed to the development of the 4D Camera used in this work. This research used resources of the National Energy Research Scientific Computing Center (NERSC), a U.S. Department of Energy Office of Science User Facility located at Lawrence Berkeley National Laboratory, operated under Contract No. DE-AC02-05CH11231 using NERSC award ERCAP0020898 and ERCAP0020897. C.S., T. P., and A.Z. were supported by the U.S. Department of Energy, Office of Science, Office of Basic Energy Sciences, Materials Sciences and Engineering Division, under Contract No. DE-AC02-05-CH11231, within the van der Waals Heterostructure Program (KCWF16) which provided for materials synthesis and within the Nanomachines Program (KC1203) which provided for TEM sample preparation. P.P. acknowledges support by the Strobe STC research center, Grant No. DMR 1548924, and an EAM Starting Grant. CH was partially supported by the National Science Foundation Graduate Research Fellowship (NSF GRFP) under Grant No. DGE-1656518, partially sponsored by the Office of Naval Research (ONR) for Hyperviper: Broadband Hyperspectral Imaging System under Award No. N00014-21-1-2788, and partially supported by University of Texas at Austin/Office of Naval Research (ONR) for the project Extraordinary Electronic Switching of Thermal Transport under Award No. UTA21-000335. V.C. acknowledges support from the ARCS Fellowship.

## Author contributions

J.M., S.S. and P.E. performed 4D STEM experiments. J.M. post-processed experimental data and performed STEM simulations. A.R. performed DFT calculations. C.S., T.P., M.J. and V.C. fabricated samples. P.E., P.P., S.S., C.H., X.X. and H.L. helped with data interpretation and writing the manuscript. The work was supervised by A.M., R.R., A.Z., P.E., N.A., K.S. and E.P.

## Competing interests

The authors declare no competing interests.
