## [Peer Review File · Nature Communications]

Imaging the electron charge density in monolayer MoS₂ at the Ångstrom scaleREVIEWER COMMENTS

Reviewer #1 (Remarks to the Author):

This paper describes attempts to separate core electrons and the valence electrons using a combination of ADF and COM 4D STEM, with a final conclusion that this is not possible with current corrected electron probe sizes.

The paper as it stands has a number of technical flaws and inconsistencies and the overall standard of English and grammar needs attention. For the latter, in the introduction "studies in literature...", p7 "outsized impact..." are examples. The use of Å and nm as units is also mixed throughout the text and figures which makes the analysis hard to follow in places.

The above comments could be fixed by careful proof reading and checks for consistency. However, there are number of more serious technical issues, as below:

1. There are other papers using phase contrast imaging compared to IAM and DFT calculations of h-BN monolayers, none of which are referenced.
2. The scales of the COM and ADF data are not normalised. The latter is plotted against a.u and this makes the subsequent subtraction at best qualitative and at worst nominal. The authors note that quantification of the ADF signal is possible, to extract fully quantitative cross sections but they surprisingly have not used this approach to quantify their ADF signal. Without a more in depth treatment of the ADF signal it is difficult to be confident in their results.
3. I was not able to find any estimate of the true electron dose (e/nm^2) or the dwell time used for the COM and ADF data.
4. There is no reference to the in house python code used for the multislice calculations which would make it hard to reproduce this work.
5. The arguments over asymmetry in the COM data are incomplete. Given the locality of the STEM probe it would be expected that the atom column would show deviation from circular symmetry in the presence of non-round aberrations but it is not obvious how the probe affects symmetry on the scale of a unit cell.
6. The argument over averaging on p11 is incomplete, unit cell and class averaging are not the same thing and I was unable to find details of how many cells were averaged or even why this is necessary.

The final conclusions are that the core electron dominate the contrast; a result that is entirely predictable from well established imaging theory in STEM. The authors also note that a smaller probe would better separate the valence and core electrons but give no indication of the probe size needed. This is important as in current generation instrument the probe size is now limited by incoherent Johnson noise for which there is no obvious instrumental solution.

Overall this paper reports a conclusion which is to a large extent obvious and together with the technical issues makes it, in my view unsuitable for publication in Nature Communications.

Reviewer #2 (Remarks to the Author):

This paper reports the current experimental limitations of direct electron charge density imaging by 4D STEM. The authors show that the probe blurring effect severely limits the imaging of valence electron charge density at atomic resolution. I feel the present findings may contain some useful information in electron microscopy community. However, the following questions should be cleared before considering this paper for publication.

1. How did the authors estimate the residual aberrations and effective source size for Fig. 3? The authors described it as 'determined empirically', but it is not easy task to uniquely determine many parameters as the authors also admitted. However, all the quantitative comparisons between experiments and simulations strongly rely on these parameters, so they should be very important.
2. Since the total charge density imaging has been already well established, it is more critical in this paper to evaluate the nuclear charge distribution. The estimated nuclear scattering factor of $Z1.9$ appear to be very high. What collection angle is used for ADF imaging? The authors

determined x for Zx via only 2 points of Mo and S, but at least three points should be required. Moreover, to quantitatively determine x , it requires quantification of ADF intensity. Did the authors perform quantitative ADF-STEM imaging, such as Ref. 26?

3. How is the matching of nuclear charge distributions between the experimentally estimated one and DFT derived one? For the nuclear charge distribution, the authors only used gaussian source size, however the authors need to consider the residual aberration effects.

4. In Fig.3, the authors estimated probe function in their experiment. How about deconvolving the probe function from the experimentally obtained charge density map? If the estimation of probe function is correct, this may give total charge density map similar to the DFT derived one.

Reviewer #3 (Remarks to the Author):

This manuscript explores the capability of 4D-STEM center of mass (CoM) imaging of electron density in monolayer MoS₂. By comparing experimental results and image simulations using DFT calculated charge density, the authors concluded that mapping charge density in MoS₂ using CoM imaging is feasible. Revealing valence electrons, however, is currently limited by probe size. The manuscript is of high interest to readers, given the rapidly rising applications of 4D-STEM and CoM imaging. The results are interesting and noteworthy. The conclusions and claims made in the manuscript are well supported. A more depth discussion on the following aspects will clarify potential confusion and improve the contribution of this work to the community:

1, the authors mentioned the achievable probe sizes at different voltages, including 80kV and 200kV. However, no discussions or simulations about how voltage influences the detectability of charge density and valence electrons were made. It would be great if the authors could estimate the optimum experimental parameters for measuring charge density and valence charges. For example, a lower voltage, such as 30kV, increases the deflection of the electron probe due to the internal electric field and enhances the detection of charge density.

2, while it is concluded that a smaller probe is needed to image charge density from valence electrons. Can the authors estimate the probe size necessary to detect valence charge density at different voltages?

3, Electron ptychography was mentioned in the manuscript. It would be very beneficial if the authors could compare the charge density map obtained by CoM and DFT (this work) with that using electron ptychography, i.e., via deconvolution of the probe function.

4, the experimental data show detectable specimen drifts, evidenced by the distortion in the hexagonal atomic arrangement and the elongation of atoms. Can the authors comment on how such sample drift impacts electric field measurements and charge density?

Our responses are in blue

Reviewer #1 (Remarks to the Author):

This paper describes attempts to separate core electrons and the valence electrons using a combination of ADF and COM 4D STEM, with a final conclusion that this is not possible with current corrected electron probe sizes.

The paper as it stands has a number of technical flaws and inconsistencies and the overall standard of English and grammar needs attention. For the latter, in the introduction “studies in literature...”, p7 “outsized impact...” are examples. The use of Å and nm as units is also mixed throughout the text and figures which makes the analysis hard to follow in places.

The above comments could be fixed by careful proofreading and checks for consistency.

We thank the reviewer for carefully reading our manuscript. We have revised the manuscript for grammar and other inconsistencies to the best of our abilities.

However, there are number of more serious technical issues, as below:

1. There are other papers using phase contrast imaging compared to IAM and DFT calculations of h-BN monolayers, none of which are referenced.

We are aware of previous work on HRTEM phase contrast imaging of hBN and have now included the following reference in our manuscript:

Meyer, J., Kurasch, S., Park, H. et al. Experimental analysis of charge redistribution due to chemical bonding by high-resolution transmission electron microscopy. Nature Mater 10, 209–215 (2011).

As we state in the results section of our paper (with reference to figure 5), we can detect the effect of the valence electron charge density ($2.5e^{-}/\text{Å}^2$) since our noise is on the order of $0.4e^{-}/\text{Å}^2$. However, we show that the electron probe significantly blurs the valence electron charge density and reduces its contrast, thus making imaging the structure of valence electron orbitals currently infeasible using the COM method.

2. The scales of the COM and ADF data are not normalised. The latter is plotted against a.u and this makes the subsequent subtraction at best qualitative and at worst nominal. The authors note that quantification of the ADF signal is possible, to extract fully quantitative cross sections but they surprisingly have not used this approach to quantify their ADF signal. Without a more in depth treatment of the ADF signal it is difficult to be confident in their results.

We note that our CoM data have intensity scale bars in milli-radians (mrad), making it quantitative. The raw ADF-STEM data is not directly used in the final valence electron density calculation. There are several reasons for this. Firstly, the HAADF detector and 4D Camera are at very different camera lengths making it difficult to get sufficient signal for a quantitative

measurement in the ADF signal. Thus, our ADF-STEM image is only used to confirm the relative signal intensities of each atom position and confirm that the intensities correspond to those expected from Mo/S atoms. To improve this comparison, we have now included multislice simulations (Supplementary Note 1 and Fig. S5) confirming that the observed ADF-peak intensities are in line with the expected scattering from Mo and S. Since we are imaging CVD grown monolayers of MoS₂, as opposed to an unknown material and thickness, full quantification of the ADF-STEM intensities is not necessary. In other words, we already know that Mo has 42 protons and S has 16 protons in its nucleus. We thus use this prior knowledge (verified by the qualitative experimental confirmation) to assign each atom position an atom type and nuclear charge. This discrete nuclear charge, placed at the centers of the experimentally determined atom sites, is then convolved with the probe shape before being subtracted from the net charge density image. We thus mentioned in the original submission that the raw ADF intensities could be used instead to show that our technique has broader applicability beyond samples with known structure and thickness.

3. I was not able to find any estimate of the true electron dose (e/nm²) or the dwell time used for the COM and ADF data.

The probe current (20 pA) and frame rate (87000 Hz) is listed in the methods section of the manuscript. We acquired a scan with 512x512 probe positions. At each position 4 detector frames were acquired in rapid succession. The electron dose D per frame can be calculated as

$$D \sim 20 \text{ pA} * (1/87000) \text{ s} * 4 \sim 5000 \text{ e-/pixel}$$

The pixel size is about 7 pm, giving us a true dose of $5000/0.07^2 \sim 1e6 \text{ e-/A}^2$. This has been added to the methods section.

4. There is no reference to the in house python code used for the multislice calculations which would make it hard to reproduce this work.

Our code calculates the shape of the electron probe which is then convolved with the projected potential or charge density predicted by DFT. The shape of the probe is calculated as outlined in the publication by E. Kirkland (*Advanced Computing in Electron Microscopy, Springer 2010*) and E. Kirkland (*Ultramicroscopy Vol 111, Issue 11, p1523-1530, 2011*). We plan to share the code on request with interested readers. HAADF-STEM multislice simulations were done using abTEM which is publicly available.

5. The arguments over asymmetry in the COM data are incomplete. Given the locality of the STEM probe it would be expected that the atom column would show deviation from circular symmetry in the presence of non-round aberrations but it is not obvious how the probe affects symmetry on the scale of a unit cell.

The probe tails due to chromatic and residual geometrical aberrations extend far beyond the FWHM of the probe and are on the order of the size of the unit cell as shown in Fig 3 of the

manuscript. Therefore, the extended probe shape can affect the charge distribution intensity on the scale of the unit cell. Simulations in Figure 3 show the effect of these tails. The highly local atom shape will mostly be affected by the mostly symmetric peak at the center.

6. The argument over averaging on p11 is incomplete, unit cell and class averaging are not the same thing and I was unable to find details of how many cells were averaged or even why this is necessary.

We thank the reviewer for pointing this out. In the original manuscript we incorrectly used the word “class” in parentheses on page 11. We indeed did not do any “class” averaging and only used “unit cell” averaging. On page 5, we report that ~25 unit cells were averaged. To be more precise, in the revised manuscript, we average larger “super-cells” which comprise ~3 hexagonal unit cells. We removed the word “class” and replaced it with “super-cells”.

Unit cell (or super-cell) averaging is required in this case because MoS₂ damages easily at the accelerating voltage used (80 kV) and cannot be imaged with the dose required to determine high quality COM measurements for every single scan position. We tested several sets of experimental parameters to determine the maximum dose we could apply. In this proof-of-principle experiment, the electron density distribution should be the same for every unit cell of MoS₂. We can thus dose fractionation across many unit cells and average their COM measurement to improve the SNR sufficiently for our measurements.

The final conclusions are that the core electrons dominate the contrast; a result that is entirely predictable from well established imaging theory in STEM. The authors also note that a smaller probe would better separate the valence and core electrons but give no indication of the probe size needed. This is important as in current generation instrument the probe size is now limited by incoherent Johnson noise for which there is no obvious instrumental solution.

In this paper, we seek to show experimentally in a well-known material (something missing in the current literature) the difficulties of directly imaging charge density. We acknowledge that it can be difficult to publish a negative result; however, we hope that our results and simulations can be used to spur further development in S/TEM aberration correction and other hardware development, along with post-processing probe deconvolution methods. Further hardware development could be made difficult due to the view by many researchers who consider the “resolution problem” solved. We show here a potentially important application of much smaller probe sizes. Larger liner tubes (*Stephan Uhlemann et al. Phys. Rev. Lett.* **111**, 2013), cooled / superconducting lenses, improved sources, and monochromation could all be needed to achieve this important goal. We have now added simulations (see supplementary figure 6) showing probe size v/s valence and core electron images. As can be seen from the figure, an improvement in resolution by a (modest) factor of 2 along with reducing probe tails due to chromatic aberrations goes a long way in separating the charge density of valence and core electrons, the latter being mostly confined to atomic sites. The ability to directly measure the electron charge density would have a very positive effect on the understanding of materials properties.

Overall this paper reports a conclusion which is to a large extent obvious and together with the technical issues makes it, in my view unsuitable for publication in Nature Communications.

We politely disagree with the reviewer that the conclusion is obvious. Right now, there is a debate in the community as to whether or not we can image the valence electron density at the atomic level using 4D-STEM. This work provides real experimental evidence from a well-known material to prove quantitatively that with current instrumentation (including uncertainties in probe correction) imaging valence electron orbitals is infeasible, but with sufficiently small probes and/or robust deconvolution techniques we may succeed in directly imaging the valence electron density.

We have not found a quantitative study in the literature showing the effect of probe blurring on the valence electron charge density. In our work, we quantify the effect of probe-induced blurring and demonstrate its effect on the valence and core electrons. Furthermore, we show that the residual and chromatic aberrations that elongate the probe tails play a really important role in interpreting charge densities. All of these aspects make our work uniquely valuable in a rapidly growing field.

Reviewer #2 (Remarks to the Author):

This paper reports the current experimental limitations of direct electron charge density imaging by 4D STEM. The authors show that the probe blurring effect severely limits the imaging of valence electron charge density at atomic resolution. I feel the present findings may contain some useful information in the electron microscopy community. However, the following questions should be cleared before considering this paper for publication.

1. How did the authors estimate the residual aberrations and effective source size for Fig. 3? The authors described it as 'determined empirically', but it is not easy task to uniquely determine many parameters as the authors also admitted. However, all the quantitative comparisons between experiments and simulations strongly rely on these parameters, so they should be very important.

The reviewer's excellent question led us to further investigate all possible contributions to probe blurring, and we found that at 80 kV chromatic aberration (~7.5 nm focal spread) is the most likely reason for the extended probe tails. As such, we have updated the manuscript with new data, simulations, and analysis to reflect this new finding. All other aberrations are minimal and well within the error bars of a corrected STEM probe accounting for some drift during operation.

2. Since the total charge density imaging has been already well established, it is more critical in this paper to evaluate the nuclear charge distribution. The estimated nuclear scattering factor of Z^{1.9} appear to be very high. What collection angle is used for ADF imaging? The authors determined χ for Z_x via only 2 points of Mo and S, but at least three points should be required. Moreover, to quantitatively determine χ , it requires quantification of ADF intensity. Did the authors perform quantitative ADF-STEM imaging, such as Ref. 26?

Our manuscript has been updated with new data showing a Z^{1.7} dependence for our collection geometry. The collection angle of our HAADF detector is relatively large due to the large physical distance between our HAADF detector (above the TEM screen) and pixelated detector (far below the screen). We believe this to be the reason for the Z^{1.7} contrast dependence. Multislice simulations (Supplementary Note 1 and Fig. S7) confirm the Mo/S contrast observed in our experiments.

We note that the ADF-STEM image intensity is used to only confirm that the relative intensities match the relative scattering expected from Mo/S. Then, since we are imaging single-layer CVD grown MoS₂ (as opposed to an unknown sample composition and thickness) we assign atom positions a known nuclear charge. Mo atoms are assigned a nuclear charge of 42 protons and S are assigned 16 protons. We then simulate the nuclear charge density by convolving our probe shape with this charge distribution. The experimental ADF-STEM intensity could be used for an unknown sample following steps reported in the quantitative STEM literature [*Phys. Rev. Lett.* **100**, 206101, 2008].

3. How is the matching of nuclear charge distributions between the experimentally estimated one and DFT derived one? For the nuclear charge distribution, the authors only used gaussian source size, however the authors need to consider the residual aberration effects.

Since the number of protons on Mo and S are already known, we simulate the nuclear charge distribution by convolving discrete nuclear charges (delta functions) at each experimentally determined atom position with the expected probe shape broadened by a 0.7 Å Gaussian source size and residual geometrical and chromatic aberrations. See Supplementary Note 1 and Fig. S7 for the details.

4. In Fig.3, the authors estimated probe function in their experiment. How about deconvolving the probe function from the experimentally obtained charge density map? If the estimation of probe function is correct, this may give total charge density map similar to the DFT derived one.

Deconvolution by phase contrast techniques (such as via ptychography) has not yet proven to give quantitatively accurate fields at atomic resolution as detailed in the discussion section of the manuscript. Ptychography was shown to improve image point resolution by super-resolution methods, but the quantitative retrieval of phase shift applied to the beam is an outstanding question in this field. We are currently exploring ptychography and other various deconvolution techniques; however, it is beyond the scope of our current work. We seek to show the current limitations and future capabilities of the relatively simple CoM measurement technique (not phase contrast STEM such as DPC or ptychography) to measure the phase shift of atomic columns.

Reviewer #3 (Remarks to the Author):

This manuscript explores the capability of 4D-STEM center of mass (CoM) imaging of electron density in monolayer MoS₂. By comparing experimental results and image simulations using DFT calculated charge density, the authors concluded that mapping charge density in MoS₂ using CoM imaging is feasible. Revealing valence electrons, however, is currently limited by probe size. The manuscript is of high interest to readers, given the rapidly rising applications of 4D-STEM and CoM imaging. The results are interesting and noteworthy. The conclusions and claims made in the manuscript are well supported. A more depth discussion on the following aspects will clarify potential confusion and improve the contribution of this work to the community:

1, the authors mentioned the achievable probe sizes at different voltages, including 80kV and 200kV. However, no discussions or simulations about how voltage influences the detectability of charge density and valence electrons were made. It would be great if the authors could estimate the optimum experimental parameters for measuring charge density and valence charges. For example, a lower voltage, such as 30kV, increases the deflection of the electron probe due to the internal electric field and enhances the detection of charge density.

We thank the reviewer for their interest in our work and this suggestion for further improvement. As the reviewer rightly points out, the voltage of the electron beam determines how small an electric field (or charge density) can be detected for a given electron dose. While it is true that a lower voltage means a higher charge density resolution (in units of $e/\text{Å}^2$), current generation microscopes are more limited by probe-induced blurring (spatial resolution) than charge density resolution. As we discuss in the results section of our manuscript, the charge density resolution of CoM we achieved ($0.4 e/\text{Å}^2$) is sufficient to detect valence electrons (which are on the order of $2.5 e/\text{Å}^2$). However, probe-induced blurring significantly smooths out the spatial distribution of valence electrons, making them an almost uniform background. A lower primary voltage like 30 keV typically leads to a larger probe size due to increased wavelength and the increased effect of chromatic aberration when compared to higher voltages. Thus, it is unlikely to be advantageous to use 30 keV without further improvements in geometrical and chromatic aberration to reach deep sub-Angstrom probe sizes at very low accelerating voltages.

2, while it is concluded that a smaller probe is needed to image charge density from valence electrons. Can the authors estimate the probe size necessary to detect valence charge density at different voltages?

We thank the reviewer for this interesting and important question. Indeed, reviewer 1 asked the same question and further details can be found in our response above. Briefly, supplementary figure 6 was added to show the effect of probe size (with no parasitic aberrations) on the valence and core electron density. As can be seen, an improvement by a factor of 2 is necessary to begin to separate valence and core electron charge densities.

3, Electron ptychography was mentioned in the manuscript. It would be very beneficial if the authors could compare the charge density map obtained by CoM and DFT (this work) with that

using electron ptychography, i.e., via deconvolution of the probe function.

As elaborated on in the discussion section of our manuscript, while ptychography can theoretically achieve super-resolution imaging (an improvement in point resolution), it has not yet been shown that it can accurately quantitatively reconstruct projected fields of an atomic lattice. This limitation of ptychography is the reason we chose to use the more direct CoM technique. We (and many others) are exploring quantitative phase contrast techniques and deem it to be outside the scope of our current manuscript.

4, the experimental data show detectable specimen drifts, evidenced by the distortion in the hexagonal atomic arrangement and the elongation of atoms. Can the authors comment on how such sample drift impacts electric field measurements and charge density?

We thank the reviewer for the attention to detail in reading our manuscript. We attribute the distortion and elongation in the charge density image to residual probe aberrations, the reason being - while sample drift can change local intensities, we do not expect it to distort the charge density on the scale of the unit cell. Probe tails due to residual aberrations, however, can extend out to many Angstroms beyond the central FWHM and can distort the charge density image across the unit cell. The ADF-STEM image in Fig 1 also supports this hypothesis - the atoms appear round and the unit cell is mostly undistorted, suggesting minimal sample drift. Further, our data set of over 1 million frames was acquired in approximately 15 seconds (87,000 Hz detector readout) reducing the effect of sample drift.

Reviewer #1 (Remarks to the Author):

The authors, in their revised manuscript have addressed many of my previous comments and, in view this has improved the paper.

My comments on their rebuttal are given below.

1. There are other papers using phase contrast imaging compared to IAM and DFT calculations of h-BN monolayers, none of which are referenced.

We are aware of previous work on HRTEM phase contrast imaging of hBN and have now included the following reference in our manuscript:

Meyer, J., Kurasch, S., Park, H. et al. Experimental analysis of charge redistribution due to chemical bonding by high-resolution transmission electron microscopy. Nature Mater 10, 209–215 (2011).

As we state in the results section of our paper (with reference to figure 5), we can detect the effect of the valence electron charge density ($2.5e^-/\text{Å}^2$) since our noise is on the order of $0.4 e^-/\text{Å}^2$. However, we show that the electron probe significantly blurs the valence electron charge density and reduces its contrast, thus making imaging the structure of valence electron orbitals currently infeasible using the COM method.

Thank you – this addresses this point.

2. The scales of the COM and ADF data are not normalised. The latter is plotted against a.u and this makes the subsequent subtraction at best qualitative and at worst nominal. The authors note that quantification of the ADF signal is possible, to extract fully quantitative cross sections but they surprisingly have not used this approach to quantify their ADF signal. Without a more in depth treatment of the ADF signal it is difficult to be confident in their results.

We note that our CoM data have intensity scale bars in milli-radians (mrad), making it quantitative. The raw ADF-STEM data is not directly used in the final valence electron density calculation. There are several reasons for this. Firstly, the HAADF detector and 4D Camera are at very different camera lengths making it difficult to get sufficient signal for a quantitative measurement in the ADF signal. Thus, our ADF-STEM image is only used to confirm the relative signal intensities of each atom position and confirm that the intensities correspond to those expected from Mo/S atoms. To improve this comparison, we have now included multislice simulations (Supplementary Note 1 and Fig. S5) confirming that the observed ADF-peak intensities are in line with the expected scattering from Mo and S. Since we are imaging CVD grown monolayers of MoS₂, as opposed to an unknown material and thickness, full quantification of the ADF-STEM intensities is not necessary. In other words, we already know that Mo has 42 protons and S has 16 protons in its nucleus. We thus use this prior knowledge (verified by the qualitative experimental confirmation) to assign each atom position an atom type and nuclear charge. This discrete nuclear charge, placed at the centers of the experimentally determined atom sites, is then convolved with the probe shape before being subtracted from the net charge density image. We thus mentioned in the original submission that the raw ADF intensities could be used instead to show that our technique has broader applicability beyond samples with known structure and thickness.

This explanation and additional material is a great help. However the sentence “*To isolate the electron charge density experimentally, we subtract the nuclear charge density determined using the ADF-STEM image from the total charge density derived using 4D-STEM CoM imaging.*” Could still lead to confusion – the method as described does not use the ADF STEM image to determine the nuclear charge but only to locate the atom sites and confirm their type from

intensity measurements. This could therefore usefully be rephrased. The finding using simulations that the core electrons dominate the Net charge density is surely obvious from figure 4 ? where the core electrons have a scale 0-1000 e/A² whereas the valence electrons have a scale 0-12 e/A². I was also unable to follow why the scale in Fig 4(c) to 4(d) changes as the difference is only a probe convolution.

3. I was not able to find any estimate of the true electron dose (e/nm²) or the dwell time used for the COM and ADF data.

The probe current (20 pA) and frame rate (87000 Hz) is listed in the methods section of the manuscript. We acquired a scan with 512x512 probe positions. At each position 4 detector frames were acquired in rapid succession. The electron dose D per frame can be calculated as $D \sim 20 \text{ pA} * (1/87000) \text{ s} * 4 \sim 5000 \text{ e-/pixel}$ The pixel size is about 7 pm, giving us a true dose of $5000/0.07^2 \sim 1e6 \text{ e-/A}^2$. This has been added to the methods section.

Thank you – this addresses this point.

4. There is no reference to the in house python code used for the multislice calculations which would make it hard to reproduce this work.

Our code calculates the shape of the electron probe which is then convolved with the projected potential or charge density predicted by DFT. The shape of the probe is calculated as outlined in the publication by E. Kirkland (*Advanced Computing in Electron Microscopy, Springer 2010*) and E. Kirkland (*Ultramicroscopy Vol 111, Issue 11, p1523-1530, 2011*). We plan to share the code on request with interested readers. HAADF-STEM multislice simulations were done using abTEM which is publicly available.

Thank you, although I still feel that a persistent link rather than code on request would be preferable.

5. The arguments over asymmetry in the COM data are incomplete. Given the locality of the STEM probe it would be expected that the atom column would show deviation from circular symmetry in the presence of non-round aberrations but it is not obvious how the probe affects symmetry on the scale of a unit cell.

The probe tails due to chromatic and residual geometrical aberrations extend far beyond the FWHM of the probe and are on the order of the size of the unit cell as shown in Fig 3 of the manuscript. Therefore, the extended probe shape can affect the charge distribution intensity on the scale of the unit cell. Simulations in Figure 3 show the effect of these tails. The highly local atom shape will mostly be affected by the mostly symmetric peak at the center.

I'm afraid I still don't agree here. The intensity in the probe at a distance comparable to the unit cell size is very small and would not be expected to have a significant effect. Moreover, the symmetry of the probe centre is exactly the same as in the tails for the same aberrations.

6. The argument over averaging on p11 is incomplete, unit cell and class averaging are not the same thing and I was unable to find details of how many cells were averaged or even why this is necessary.

We thank the reviewer for pointing this out. In the original manuscript we incorrectly used the word "class" in parentheses on page 11. We indeed did not do any "class" averaging and only used "unit cell" averaging. On page 5, we report that ~25 unit cells were averaged. To be more precise, in the revised manuscript, we average larger "super-cells" which comprise ~3 hexagonal unit cells. We removed the word "class" and replaced it with "super-cells".

Thank you – this addresses this point.

Unit cell (or super-cell) averaging is required in this case because MoS₂ damages easily at the accelerating voltage used (80 kV) and cannot be imaged with the dose required to determine high quality COM measurements for every single scan position. We tested several sets of experimental parameters to determine the maximum dose we could apply. In this proof-of-principle experiment, the electron density distribution should be the same for every unit cell of MoS₂. We can thus dose fractionation across many unit cells and average their COM measurement to improve the SNR sufficiently for our measurements.

Surely this argument about damage means that the final charge density is not that of pristine MoS₂ ?

7. The final conclusions are that the core electron dominate the contrast; a result that is entirely predictable from well established imaging theory in STEM. The authors also note that a smaller probe would better separate the valence and core electrons but give no indication of the probe size needed. This is important as in current generation instrument the probe size is now limited by incoherent Johnson noise for which there is no obvious instrumental solution.

In this paper, we seek to show experimentally in a well-known material (something missing in the current literature) the difficulties of directly imaging charge density. We acknowledge that it can be difficult to publish a negative result; however, we hope that our results and simulations can be used to spur further development in S/TEM aberration correction and other hardware development, along with post-processing probe deconvolution methods. Further hardware development could be made difficult due to the view by many researchers who consider the “resolution problem” solved. We show here a potentially important application of much smaller probe sizes. Larger liner tubes (*Stephan Uhlemann et al. Phys. Rev. Lett. 111, 2013*), cooled / superconducting lenses, improved sources, and monochromation could all be needed to achieve this important goal. We have now added simulations (see supplementary figure 6) showing probe size v/s valence and core electron images. As can be seen from the figure, an improvement in resolution by a (modest) factor of 2 along with reducing probe tails due to chromatic aberrations goes a long way in separating the charge density of valence and core electrons, the latter being mostly confined to atomic sites. The ability to directly measure the electron charge density would have a very positive effect on the understanding of materials properties.

I accept the argument about publishing negative results and that measurement of charge densities is important. However, I disagree with the view that the “resolution problem is solved”. The issue is that there are no obvious routes to achieving the required factor of 2 in resolution. This section would need rewriting to be acceptable.

8. Overall this paper reports a conclusion which is to a large extent obvious and together with the technical issues makes it, in my view unsuitable for publication in Nature Communications.

We politely disagree with the reviewer that the conclusion is obvious. Right now, there is a debate in the community as to whether or not we can image the valence electron density at the atomic level using 4D-STEM. This work provides real experimental evidence from a well-known material to prove quantitatively that with current instrumentation (including uncertainties in probe correction) imaging valence electron orbitals is infeasible, but with sufficiently small probes and/or robust deconvolution techniques we may succeed in directly imaging the valence electron density.

We have not found a quantitative study in the literature showing the effect of probe blurring on the valence electron charge density. In our work, we quantify the effect of probe-induced blurring and demonstrate its effect on the valence and core electrons. Furthermore, we show that the residual and chromatic aberrations that elongate the probe tails play a really important role in interpreting charge densities. All of these aspects make our work uniquely valuable in a rapidly growing field.

I accept the argument that the paper demonstrates that imaging valence electrons is feasible in theory. However as already noted there is limited prospect of the instrumentation needed being available. If the main message of the paper were quantification of probe induced blurring this might be addressed but this would then be a very specialised paper.

Reviewer #2 (Remarks to the Author):

The authors considered my previous comments very carefully and properly revised the manuscript with several additional data. I can now recommend this paper for publication.

Reviewer #4 (Remarks to the Author):

The paper discusses the limitations of 4DSTEM for measuring the electronic properties of 2D materials. The paper has already been reviewed three times and as such the results are now coherent and most errors have been removed (there are still a few typos which I leave the authors to find).

My feeling is that although this paper gives a negative result, which for specialists in the field may be rather obvious, the results do not seem to be obvious to the wider community as a whole. I enjoyed reading the paper, their story is well constructed and easy to follow. This is a very fashionable subject and it is refreshing to read something more based on the reality.

I have at this stage no comments on the contents of the manuscript as it is, as it has already been reviewed thoroughly.

I still find it a shame that Figure 6 is in the SI as it gives solutions to the problem.

Additionally, it would be interesting to hear the authors opinion on the technique as used for more complex systems and real materials rather than sheets of 2D materials.

It is not for me to judge the suitability of the work for a Nature publication, but I find the work interesting and useful for a wide community. It is written in a way than non-physicists would also understand the content easily.

Final response to referees #1 and #4 (in brown)

Referee #1

The authors, in their revised manuscript have addressed many of my previous comments and, in view this has improved the paper.

My comments on their rebuttal are given below.

1. There are other papers using phase contrast imaging compared to IAM and DFT calculations of h-BN monolayers, none of which are referenced.

We are aware of previous work on HRTEM phase contrast imaging of hBN and have now included the following reference in our manuscript:

Meyer, J., Kurasch, S., Park, H. et al. Experimental analysis of charge redistribution due to chemical bonding by high-resolution transmission electron microscopy. Nature Mater 10, 209–215 (2011).

As we state in the results section of our paper (with reference to figure 5), we can detect the effect of the valence electron charge density ($2.5e^{-}/\text{Å}^2$) since our noise is on the order of $0.4 e^{-}/\text{Å}^2$. However, we show that the electron probe significantly blurs the valence electron charge density and reduces its contrast, thus making imaging the structure of valence electron orbitals currently infeasible using the COM method.

Thank you – this addresses this point.

2. The scales of the COM and ADF data are not normalised. The latter is plotted against a.u and this makes the subsequent subtraction at best qualitative and at worst nominal. The authors note that quantification of the ADF signal is possible, to extract fully quantitative cross sections but they surprisingly have not used this approach to quantify their ADF signal. Without a more in depth treatment of the ADF signal it is difficult to be confident in their results.

We note that our CoM data have intensity scale bars in milli-radians (mrad), making it quantitative. The raw ADF-STEM data is not directly used in the final valence electron density calculation. There are several reasons for this. Firstly, the HAADF detector and 4D Camera are at very different camera lengths making it difficult to get sufficient signal for a quantitative measurement in the ADF signal. Thus, our ADF-STEM image is only used to confirm the relative signal intensities of each atom position and confirm that the intensities correspond to those expected from Mo/S atoms. To improve this comparison, we have now included multislice

simulations (Supplementary Note 1 and Fig. S5) confirming that the observed ADF-peak intensities are in line with the expected scattering from Mo and S. Since we are imaging CVD grown monolayers of MoS₂, as opposed to an unknown material and thickness, full quantification of the ADF-STEM intensities is not necessary. In other words, we already know that Mo has 42 protons and S has 16 protons in its nucleus. We thus use this prior knowledge (verified by the qualitative experimental confirmation) to assign each atom position an atom type and nuclear charge. This discrete nuclear charge, placed at the centers of the experimentally determined atom sites, is then convolved with the probe shape before being subtracted from the net charge density image. We thus mentioned in the original submission that the raw ADF intensities could be used instead to show that our technique has broader applicability beyond samples with known structure and thickness.

This explanation and additional material is a great help. However the sentence “*To isolate the electron charge density experimentally, we subtract the nuclear charge density determined using the ADF-STEM image from the total charge density derived using 4D-STEM CoM imaging.*” Could still lead to confusion – the method as described does not use the ADF STEM image to determine the nuclear charge but only to locate the atom sites and confirm their type from intensity measurements. This could therefore usefully be rephrased. The finding using simulations that the core electrons dominate the Net charge density is surely obvious from figure 4 ? where the core electrons have a scale 0-1000 e/A² whereas the valence electrons have a scale 0-12 e/A². I was also unable to follow why the scale in Fig 4(c) to 4(d) changes as the difference is only a probe convolution.

We have rephrased this part of our manuscript as per the reviewer's suggestion. The new sentence in the 'Imaging the electron charge density' section reads

To isolate the electron charge density experimentally, the nuclear charge density needs to be subtracted from the total charge density derived using 4D-STEM CoM imaging. In our experiments, we find that the ADF-STEM intensity for S and Mo scales as $\sim Z^{1.7}$ which we validate with multislice simulations using abTEM28 (see Supplementary Note 1 and Supplementary Figure 5). Using prior knowledge that the sample is 1 layer of MoS₂, we assign each experimentally determined atom position a discrete nuclear charge'

The core electrons have a higher projected charge density simply because they are confined to smaller orbitals in the atomic core, usually $\sim 0.1\text{\AA}$. For example, if 10 electrons are confined to 0.1\AA core orbitals, their charge density is $\sim 1000\text{ e/\AA}^2$. If those same 10 electrons are instead confined to 1 \AA valence orbitals, their charge density is 10 e/\AA^2 . So the same 10 electrons can have vastly different projected charge densities simply based on the orbital size. This leads to the (reasonable) expectation that the charge density between atoms (which are usually few \AA apart in lattices) is dominated by the valence electrons, i.e. any modulation away from atomic cores should arise due to valence electrons. We show quantitatively that because of probe convolution, this is not true. Probe convolution results in the core electron charge density being spread out everywhere, even between the atoms as figure 4 shows. Therefore, although one expects only valence electrons to contribute to chemical bonds, probe convolution results in the charge density between atoms also being dominated by core electrons.

The reason why the scale bar changes is simply because probe convolution spreads the core electrons from 0.1 Å orbitals to about 1 Å (the probe size). In our calculations, the probe is normalized such that the area under the curve is equal to unity. As the total number of electrons (integral of the image) must stay constant under convolution, the apparent charge density decreases with convolution.

3. I was not able to find any estimate of the true electron dose (e/nm²) or the dwell time used for the COM and ADF data.

The probe current (20 pA) and frame rate (87000 Hz) is listed in the methods section of the manuscript. We acquired a scan with 512x512 probe positions. At each position 4 detector frames were acquired in rapid succession. The electron dose D per frame can be calculated as $D \sim 20 \text{ pA} * (1/87000) \text{ s} * 4 \sim 5000 \text{ e-/pixel}$. The pixel size is about 7 pm, giving us a true dose of $5000/0.07^2 \sim 1e6 \text{ e-/Å}^2$. This has been added to the methods section.

Thank you – this addresses this point.

4. There is no reference to the in house python code used for the multislice calculations which would make it hard to reproduce this work.

Our code calculates the shape of the electron probe which is then convolved with the projected potential or charge density predicted by DFT. The shape of the probe is calculated as outlined in the publication by E. Kirkland (*Advanced Computing in Electron Microscopy, Springer 2010*) and E. Kirkland (*Ultramicroscopy Vol 111, Issue 11, p1523-1530, 2011*). We plan to share the code on request with interested readers. HAADF-STEM multislice simulations were done using abTEM which is publicly available.

Thank you, although I still feel that a persistent link rather than code on request would be preferable.

We have now uploaded our code into a repository on Zenodo
(<https://doi.org/10.5281/zenodo.7916671>).

5. The arguments over asymmetry in the COM data are incomplete. Given the locality of the STEM probe it would be expected that the atom column would show deviation from circular symmetry in the presence of non-round aberrations but it is not obvious how the probe affects symmetry on the scale of a unit cell.

The probe tails due to chromatic and residual geometrical aberrations extend far beyond the FWHM of the probe and are on the order of the size of the unit cell as shown in Fig 3 of the manuscript. Therefore, the extended probe shape can affect the charge distribution intensity on the scale of the unit cell. Simulations in Figure 3 show the effect of these tails. The highly local atom shape will mostly be affected by the mostly symmetric peak at the center.

I'm afraid I still don't agree here. The intensity in the probe at a distance comparable to the unit cell size is very small and would not be expected to have a significant effect. Moreover, the symmetry of the probe centre is exactly the same as in the tails for the same aberrations.

We politely disagree with the reviewer's statement 'the intensity in the probe at a distance comparable to the unit cell size is very small and would not be expected to have a significant effect'. We refer to Fig 6 from Earl J Kirkland's paper 'On the Optimum Probe shape in aberration corrected ADF-STEM' (*Ultramicroscopy, Volume 111, Issue 11, pages 1523-1530*) which compares the radially integrated probe intensity for an aberration free probe v/s one with reasonable aberrations (similar to ours). The figure shows that with reasonable aberrations, only 25% of the probe intensity is within a 1 Å diameter, whereas the remaining 75% of the probe intensity is between 1 to 4 Å, which is on the order of typical unit cell sizes. Fig 5 from the same paper also shows that the symmetry of the probe tails might be different from the probe center when multiple aberrations are present.

6. The argument over averaging on p11 is incomplete, unit cell and class averaging are not the same thing and I was unable to find details of how many cells were averaged or even why this is necessary.

We thank the reviewer for pointing this out. In the original manuscript we incorrectly used the word "class" in parentheses on page 11. We indeed did not do any "class" averaging and only used "unit cell" averaging. On page 5, we report that ~25 unit cells were averaged. To be more precise, in the revised manuscript, we average larger "super-cells" which comprise ~3 hexagonal unit cells. We removed the word "class" and replaced it with "super-cells".

Thank you – this addresses this point.

Unit cell (or super-cell) averaging is required in this case because MoS₂ damages easily at the accelerating voltage used (80 kV) and cannot be imaged with the dose required to determine high quality COM measurements for every single scan position. We tested several sets of experimental parameters to determine the maximum dose we could apply. In this proof-of-principle experiment, the electron density distribution should be the same for every unit cell of MoS₂. We can thus dose fractionation across many unit cells and average their COM measurement to improve the SNR sufficiently for our measurements.

Surely this argument about damage means that the final charge density is not that of pristine MoS₂?

As mentioned in our supplementary information (Fig 3), we only pick pristine unit cells from the larger image for unit cell averaging, leaving out defects and other damaged areas. This ensures that our final charge density image comes from pristine parts of the sample that were not damaged during the scan.

7. The final conclusions are that the core electrons dominate the contrast; a result that is entirely predictable from well established imaging theory in STEM. The authors also note that a smaller probe would better separate the valence and core electrons but give no indication of the probe size needed. This is important as in current generation instruments the probe size is now limited by incoherent Johnson noise for which there is no obvious instrumental solution.

In this paper, we seek to show experimentally in a well-known material (something missing in the current literature) the difficulties of directly imaging charge density. We acknowledge that it can be difficult to publish a negative result; however, we hope that our results and simulations can be used to spur further development in S/TEM aberration correction and other hardware development, along with post-processing probe deconvolution methods. Further hardware development could be made difficult due to the view by many researchers who consider the “resolution problem” solved. We show here a potentially important application of much smaller probe sizes. Larger linetubes (*Stephan Uhlemann et al. Phys. Rev. Lett.* **111**, 2013), cooled / superconducting lenses, improved sources, and monochromation could all be needed to achieve this important goal. We have now added simulations (see supplementary figure 6) showing probe size v/s valence and core electron images. As can be seen from the figure, an improvement in resolution by a (modest) factor of 2 along with reducing probe tails due to chromatic aberrations goes a long way in separating the charge density of valence and core electrons, the latter being mostly confined to atomic sites. The ability to directly measure the electron charge density would have a very positive effect on the understanding of materials properties.

I accept the argument about publishing negative results and that measurement of charge densities is important. However, I disagree with the view that the “resolution problem is solved”. The issue is that there are no obvious routes to achieving the required factor of 2 in resolution. This section would need rewriting to be acceptable.

The routes to achieve smaller probe sizes are indeed non-trivial, but by no means impossible. CEOS recently demonstrated spherical aberration correction up to a 70 mrad aperture angle (*S. Uhlemann et al, Microsc. Microanal.* **28 (Suppl 1), 2022), although their final probe size was limited by chromatic aberrations. The other route to better resolution is improved post processing probe deconvolution methods such as ptychography, whose quantitative robustness is yet to be demonstrated. We have rewritten our discussion section elaborating on all these points, with appropriate references.**

8. Overall this paper reports a conclusion which is to a large extent obvious and together with the technical issues makes it, in my view unsuitable for publication in Nature Communications.

We politely disagree with the reviewer that the conclusion is obvious. Right now, there is a debate in the community as to whether or not we can image the valence electron density at the atomic level using 4D-STEM. This work provides real experimental evidence from a well-known material to prove quantitatively that with current instrumentation (including uncertainties in probe correction) imaging valence electron orbitals is infeasible, but with sufficiently small probes and/or robust deconvolution techniques we may succeed in directly imaging the valence electron density.

We have not found a quantitative study in the literature showing the effect of probe blurring on the valence electron charge density. In our work, we quantify the effect of probe-induced blurring and demonstrate its effect on the valence and core electrons. Furthermore, we show that the residual and chromatic aberrations that elongate the probe tails play a really important role in interpreting charge densities. All of these aspects make our work uniquely valuable in a rapidly growing field.

I accept the argument that the paper demonstrates that imaging valence electrons is feasible in theory. However as already noted there is limited prospect of the instrumentation needed being available. If the main message of the paper were quantification of probe induced blurring this might be addressed but this would then be a very specialised paper.

We thank the reviewer for their feedback. We believe that our work is uniquely valuable as it:

- (1) Shows quantitatively the limitations of current instrumentation in solving an important scientific issue (imaging valence orbitals using 4D STEM)**
- (2) Provides suggestions and motivation for future developments to address these limitations.**

We hope that future developments in both instrumentation as well as probe deconvolution (post processing) will enable imaging valence electron orbitals using 4D STEM. We now discuss this in more detail at the end of the Discussion section of our manuscript.

Referee #4

The paper discusses the limitations of 4DSTEM for measuring the electronic properties of 2D materials. The paper has already been reviewed three times and as such the results are now coherent and most errors have been removed (there are still a few typos which I leave the authors to find).

My feeling is that although this paper gives a negative result, which for specialists in the

field may be rather obvious, the results do not seem to be obvious to the wider community as a whole. I enjoyed reading the paper, their story is well constructed and easy to follow. This is a very fashionable subject and it is refreshing to read something more based on the reality.

I have at this stage no comments on the contents of the manuscript as it is, as it has already been reviewed thoroughly.

I still find it a shame that Figure 6 is in the SI as it gives solutions to the problem. Additionally, it would be interesting to hear the authors opinion on the technique as used for more complex systems and real materials rather than sheets of 2D materials.

It is not for me to judge the suitability of the work for a Nature publication, but I find the work interesting and useful for a wide community. It is written in a way than non-physicists would also understand the content easily.

We thank the reviewer for their feedback on the manuscript and its wider impact. As per their suggestion, we have now moved Supplementary Fig 6 to the main manuscript. We have also discussed the applicability of this technique to other materials in the discussion section.